# Crystalline methamphetamine (ice) use prior to youth detention: A forensic concern or a public health issue?

Benjamin Spivak[1], Stephane Shepherd[1]*, Rohan Borschmann[2,3,4,5], Stuart A. Kinner[2,3,6,7,8,9], James R. P. Ogloff[1], Henning Hachtel[1,10]

**1** Centre for Forensic Behavioural Science, Swinburne University of Technology, Level 1, Alphington, Victoria, Australia, **2** Centre for Adolescent Health, Murdoch Children's Research Institute; Melbourne, Australia, **3** Centre for Health Equity, Melbourne School of Population and Global Health; The University of Melbourne, Melbourne, Australia, **4** Department of Psychiatry; The University of Melbourne, Melbourne, Australia, **5** Section for Women's Mental Health, Health Service and Population Research Department; Institute of Psychiatry, Psychology & Neuroscience; King's College London, London, United Kingdom, **6** Mater Research Institute-UQ, University of Queensland, Brisbane, Australia, **7** Griffith Criminology Institute, Griffith University, Brisbane, Australia, **8** School of Public Health and Preventive Medicine, Monash University, Melbourne, Australia, **9** Netherlands Institute for the Study of Crime and Law Enforcement, Amsterdam, Netherlands, **10** Universitäre Psychiatrische Kliniken (UPK), University of Basel, Basel, Switzerland

* sshepherd@swin.edu.au

**Data Availability Statement:** Data cannot be made available due to privacy concerns around sensitive information (criminal records) pertaining to young people at this time. The conditions of the existing

## Abstract

Links between crystalline methamphetamine (CM) use and criminal offending are often drawn in the media; however, there has been little scientific research into this relationship. The aim of this study was to ascertain the prevalence and correlates of lifetime CM use among a sample of young people in detention in Australia and to examine whether an association exists between lifetime CM use and recidivism in this population. The sample included 202 young people (164 males) in youth detention in the state of Victoria, Australia. Participants were administered questionnaires related to lifetime substance use and socio-environmental experiences. Lifetime mental health data and offending data were obtained for each participant from public mental health and policing databases. More than one third (38%) of the sample reported lifetime CM use. In multivariate logistic regression analyses, older age, male gender, polysubstance use, and high levels of community disorganisation were associated with CM use. The presence of a psychiatric diagnosis over the lifetime was not significantly associated with CM use. CM use was also not significantly associated with violent recidivism. Efforts to address CM use and related harm in detained youth should include community-based strategies to reduce CM use among this vulnerable population following their release from detention. However, the findings suggest that CM use on its own is unlikely to be an important consideration for professionals concerned with determining which young people should be selected for treatment designed to reduce the risk of violent recidivism.

ethics agreement does not permit for data sharing. To access the data, interested parties will need to seek permission from the Victorian Department of Justice and Community Safety (Youth Justice) Human Research Ethics Committee (ethics@justice.vic.gov.au).

**Funding:** This study was funded by the Australian Research Council (www.arc.gov.au) grant DP1095697. The funder had no role in study design, data collection and analysis, decision to publish, or preparation of the manuscript.

**Competing interests:** The authors have declared that no competing interests exist.

# Introduction

Crystalline methamphetamine (CM; also known as 'crystal meth' or 'ice') is a stimulant commonly smoked or injected as a recreational drug [1,2]. As with other forms of methamphetamine use, CM use normally results in feelings of euphoria and reduced inhibitions[2]. However, it is more potent, addictive, and faster acting than other forms of methamphetamine [1]. Moreover, CM is increasingly the main form of methamphetamine used in Australia [3]. Adverse psychological effects of CM use include anxiety, aggression, paranoia, irritability, depression, violence, and psychotic symptoms[4–9].

Few large-scale prevalence studies have been undertaken to assess the prevalence of CM use in the general community. The 2016 Australian National Drug Strategy Household Survey reported that the highest proportion of people using CM were in the adolescent and young adult age categories [3]. Among Australians aged 14–19 years, approximately 0.8% reported using CM in the past 12 months in 2016; this increased to 3.0% for young people aged 20–29 years [3]. Less information is available about the prevalence of lifetime CM use although estimates of up to 4.0% have been reported for young people aged 15 to 29 years in Australia [10]. More than half of methamphetamine users in Australia use CM [3]. CM users are also more likely to use on a regular basis compared with users of other forms of methamphetamine [3].

Adolescence is an important phase of neuro-development, with the brain still maturing and developing [11,12]. For this reason, and the deleterious psychological effects of CM use, CM use during adolescence may have serious long-term psychiatric and neurological consequences. A number of studies have investigated the correlates of adolescent methamphetamine use. Associations have been reported between adolescent methamphetamine use and psychiatric symptomatology [13–16], antisocial behaviour [15–17], risky sexual practices [17], polysubstance use [16], interaction with antisocial peers [14,17], peer drug use [14,17], and violence perpetration [16].

Links between CM use and criminal offending are often drawn in the media [18–20]; however, there has been little scientific research into this relationship. Again, the research has focused on methamphetamine use more broadly. Several studies have found that young methamphetamine users commit both violent and non-violent criminal offences, with lifetime prevalence estimates ranging from 35%–55% of users committing crime [21–24]. However, after reviewing these studies, McKetin et al. [25] speculated that the rates of involvement in violent crime for methamphetamine users are likely similar to rates of violent crime among users of other substances.

To our knowledge, only one study has specifically examined the relationship between CM use and violent recidivism in adolescence. Iritani et al. [4] examined the association between CM use and criminal charges for 14,322 young adults in the United States. After controlling for socio-demographic variables and the use of other substances, the authors found that the association between CM use and criminal charges was not significant.

Given recent increases in CM use among young people in Australia, there is a clear need for further research exploring the relationship between CM use and violence in adolescent populations. This is especially true for at-risk or vulnerable groups such as young people in juvenile detention. Several studies have reported higher rates of substance use [26–28] and psychiatric symptomatology [29–32] for young people in detention relative to the age-matched general population. However, no studies to date have investigated CM use specifically among young people in detention. If CM use is associated with repeat violent recidivism, then this has ramifications for risk management and reducing recidivism. CM use in this population is also a significant health concern, given the association between CM use and adverse psychiatric and health outcomes [3–6]. This study aimed to address this substantial gap in knowledge by

seeking to identify 1) the lifetime prevalence of CM use, 2) the correlates of CM use, and 3) the association between CM use and future violent criminal offending, among a sample of young people in detention.

## Materials and methods

### Settings

Participants were recruited from both youth justice centres in Victoria, Australia. The first, Parkville Youth Justice Precinct (PYJP), accommodates males and females aged from 10 to 17 years, and young women aged 18 to 20 years, who have been remanded or sentenced in a Victorian court. The second, Malmsbury Youth Justice Centre (MYJC), accommodates young men aged 18 to 20 years who have been remanded or sentenced in a Victorian court.

### Procedure

Participants were eligible to participate if they were English speaking and able to comprehend the participatory explanation form. Detainees were not approached if Justice Centre staff deemed them to be of unstable mood or likely to exhibit extreme aggressive behavior if interviewed. Consenting participants were then administered a semi-structured interview in custody by masters level clinical psychology researchers. The interview canvassed prior self-reported delinquent behaviour and initiation into criminal activity, peer/family relationships, attitudes towards offending, traumatic experiences, mental health symptoms, and education/employment involvement. The interview responses were used to code the Structured Assessment of Violence Risk in Youth (SAVRY) risk instrument (described below). Participants were interviewed individually in private rooms allocated by youth justice custodial centre staff. The duration of each assessment was approximately 90 minutes.

### Ethics, Consent and Permissions

Written informed consent was obtained from all participants prior to the interview commencing. Parental consent was not obtained. Consent for participants under 18 years of age fell within the "mature minor" concept as described in Australian case law [33], where mental competency is determined by the ability of an underage participant to understand or appreciate information pertaining to their partaking in, and the nature of, the study. No participants were determined to lack the capacity to provide informed consent and no participants refused to provide informed consent. The study was approved by the Victorian Department of Human Services and the Monash University Human Research Ethics Committee (CF10/2960–2010001629).

### Measures

Measures of CM use, other substance use, past psychiatric diagnosis, and the Structured Assessment of Violence Risk in Youth (SAVRY), were administered at baseline. Violent recidivism during the subsequent 18 months was assessed through prospective linkage with statewide police records.

### Crystalline methamphetamine (CM) use

Participants were asked if they had ever used CM. Colloquial Australian terms for CM such as 'ice' or 'shards' were included in the questioning. Responses were coded into a binary variable (0 = No CM use; 1 = lifetime CM use).

## Other substance use

The Kiddie Schedule for Affective Disorders and Schizophrenia Present—Lifetime Version (KSADS-PL) is a semi-structured interview schedule designed to screen for a range of psychological disorders in children and adolescents [34]. It comprises six sections and five diagnostic supplements. Diagnostic supplement 5 –Substance Abuse and Other Disorders–was used in the present study. This supplement contains questions concerning lifetime use of a number of illicit substances, and frequency of alcohol use per month. Each item relating to lifetime use of an illicit substance is scored from 0–2 (0 indicating 'no information', 1 indicating 'no use', and 2 indicating 'substance use'). Frequency of alcohol use is scored by recording the number of days per month a respondent reports using alcohol on average. The KSADS-PL Diagnostic supplement 5 items used in the present study covered lifetime use of cannabis, sedatives/hypnotics/anxiolytics, opioids, hallucinogens, cocaine and solvents/inhalants; and frequency of alcohol use per month.

## Psychiatric diagnosis

Data on the presence of any past psychiatric diagnosis was obtained from the state-wide public mental health database, known as the Redevelopment of Acute and Psychiatric Information Directions (RAPID). Diagnoses are recorded by health workers in the public mental health system through the client management interface under mandatory requirements specified under the Australian Health Care Agreement [35]. A diagnosis was considered present if a participant had a registered diagnosis of anxiety disorder (ICD codes: F41.1, F41.9, F42.0, F93.0), mood disorder (ICD-10 codes: F31.1, F31.8, F33, F34.1), schizophrenia (ICD-10 codes: F20.9, F21, F25.9), psychosis (ICD-10 codes: F29, F10.95, F11.95, F12.95, F13.95, F14.95, F15.95, F16.95, F18.95, F19.95), trauma associated disorder (ICD-10 codes: F43.10, F43.2, F43.8), personality disorder (ICD-10 codes: F60.2, F60.3, F60.9), behavioural disorder (F63, F91.3, F91.9,), or a neuro-developmental disorder (ICD codes: F70, F71, F72, F73, F78, F79, F81.9, F89, F90.9, F299.0, Q86.0). Given the low proportion of participants across diagnostic categories, a decision was made to create a binary variable indicating the presence of any diagnosis (0 = No diagnosis present; 1 = Any diagnosis present)

## Structured Assessment of Violence Risk in Youth (SAVRY)

The SAVRY is a structured professional judgment instrument designed to assess risk for violence in young people aged 12–18 years [36]. It comprises 24 risk items across three subscales assessing Historical, Socio/Contextual, and Individual domains. Each SAVRY risk item is coded on a three-point scale (0 = Low, 1 = Moderate, 2 = High) to indicate the presence and severity of the risk item.

For the purposes of the present study, four SAVRY items that had shown previous empirical and/or theoretical relationships with CM use were selected to test for an association between CM use and violence risk in the sample. These were:

a) Lifetime history of violence (Low = no acts of violence, Moderate = 1 to 2 acts of violence; High = 3 or more acts of violence);

b) peer delinquency (Low = Youth does not associate with delinquent peers, Moderate = Youth occasionally associates with other delinquents or criminals or regularly associates with other youth who have engaged in relatively infrequent or minor antisocial acts, High = Youth frequently associates with other delinquents or criminals, including other youth who regularly engage in antisocial acts and/or youth is involved with gang activities or is a gang member);

c) community disorganisation (Low = Youth lives in a community with low rates of crime, poverty and violence, Moderate = Youth lives in a community with some problems related to higher rates of crime, poverty, and/or violence, High = Youth lives in a community with significant problems relating to high rates of crime, poverty, and/or violence);

d) Poor parental management (Low = Youth is receiving consistent and appropriate parental discipline, adequate supervision and involvement by parents, Moderate = Youth is receiving discipline that is sometimes inconsistent, but not overly strict or overly permissive on a regular basis; High = Youth is receiving discipline that is extremely inconsistent, or that is consistently overly strict or overly permissive).

## Subsequent violent recidivism

Follow-up data were collected for up to 18 months post-interview. All participants had a minimum of six months follow up time. Participants consented to allow Victoria Police to release their de-identified criminal history from Victoria Police (i.e., the Law Enforcement Assistance Program) database to the research team. In line with the World Health Organisation's definition of violence [37], we defined violent recidivism as incorporating both intentional physical harm and threats to cause physical harm.

Police charges were selected for the outcome rather than arrests or convictions. We reasoned that arrests might underestimate violent recidivism given that not all violent offences result in arrest. We also decided against using convictions given a) the likely low base rate of conviction over the limited follow up period and the likely underestimate of violent recidivism caused by decisions not to proceed to trial.

In order to create a de-identified dataset, a statistical linkage key was used which linked names and birth dates of participants with a unique code. The linkage key was stored separately from the matched dataset.

## Statistical analysis

We calculated the percentage (with 95% confidence intervals) of participants who reported using each of the following substances: CM, cannabis, sedatives/hypnotics/anxiolytics, opioids, hallucinogens, cocaine, and solvents/inhalants. In addition, a polysubstance use score was calculated by summing the number of substances that participants reported using (not including CM) and dichotomising the variable into those who had and had not used two or more substances (polysubstance use).

Multivariate logistic regression models were constructed to estimate associations between independent variables and CM use (model 1) and independent variables and violent recidivism (models 2 and 3). Given the large number of terms in each model relative to events in the outcome variable, each model was penalized to reduce overfitting and overestimates of associations [38]. Penalty terms were chosen using the Pentrace function in the RMS package [39], which determines the optimum penalty term by constructing multiple models with differing penalty terms and selecting the penalty term that results in best model fit (as determined by the Akaike information criterion). Penalty terms used in each of the models were model 1 = 0.5; model 2 = 34.7; model 3 = 10.4.

Model 1 aimed to identify correlates of CM use. Potential correlates were selected based on associations reported in previous literature [4,12–16] and included demographic variables (age and sex), substance use variables (frequency of alcohol use, and polysubstance use), and social variables (psychiatric diagnosis to baseline, community disorganization, and poor parental management). Odds ratios (ORs) and associated 95% confidence intervals (95%CI) were computed for each variable, first unadjusted and then adjusted for all other variables in the model.

The relationship between CM use and subsequent violent recidivism was then assessed through two multivariate logistic regression models. Using a single model for all variables was not possible due to the relatively small sample size. Model 2 examined the relationship between lifetime use of various substance classes (including CM) and violent recidivism. The second examined the association between CM use and violent recidivism together with polysubstance use, psychiatric diagnosis, and history of violence. For each of the significance tests performed in the current study, the alpha level was set at .05 and all tests were two-tailed.

## Results

### Sample

A total of 202 (*male* = 164 [81%]) young people consented to take part in the study. This represents a considerable recruitment faction given that population estimates suggest that within the state of Victoria, the number of young people in detention on an average night ranged from 143 to 182 from 2011 to 2013, with an average of 157 young peple in detention on an average night in the time period of recruitment[40]. The mean age of participants was 16.7 (*SD* = 1.8, range: 12–21) years. The majority (86%, *n* = 174) had previously been charged by police for a violent offence and all participants had a self-reported history of violence. The most common index offence in the cohort was serious assault (33%, *n* = 49).

### Prevalence of crystalline methamphetamine use

The prevalence of lifetime use of CM and other substances is presented in Table 1. Cannabis (86%) was the substance most commonly used in the sample, followed by CM (38%). All participants who reported using CM also reported polysubstance use. Polysubstance use was reported by 33% (n = 65) of the sample.

CM use was reported by 57% (n = 37) of participants who reported polysubstance use. Those who reported using CM reported having used, on average, one substance other than CM (median = 1, range = 0–6). The number of additional substances reported by those who had not used CM was similar (median = 1, range = 0–5). The substance most frequently co-reported with CM use was cannabis (n = 72, 95%). A small proportion of the sample (n = 20, 10%) had not used any of the substances listed.

### Correlates of crystalline methamphetamine use

A multivariate logistic model was constructed to identify correlates of CM use in the sample. Nagelkerke's $R^2$ for the model was $R^2 = 0.42$.

**Table 1. Lifetime substance use prevalence and associated 95% confidence intervals.**

| Substance | n/N | % | (95% CI) |
|---|---|---|---|
| Cannabis | 172/200 | 86 | 80, 90 |
| Crystalline Methamphetamine | 76/202 | 38 | 31, 45 |
| Sedatives/Hypnotics/Anxiolytics | 35/199 | 18 | 13, 24 |
| Opioids | 31/200 | 16 | 11, 21 |
| Hallucinogens | 19/200 | 10 | 6, 15 |
| Cocaine | 18/200 | 9 | 6, 14 |
| Solvents/Inhalants | 18/200 | 9 | 6, 14 |
| Polysubstance use[a] | 65/198 | 33 | 26, 40 |

[a]Polysubstance use did not include CM use.

Four variables were significantly associated with CM use in the adjusted model: older age, male gender, polysubstance use, and high community disorganisation (Table 2).

## Association between crystalline methamphetamine use and subsequent violent recidivism

During the follow up period, just over half of the sample were charged with a violent offence (n = 103, 51.0%), with 52 (25.7%) being charged within two months of release.

The association between CM use and violent recidivism was examined in two multivariate logistic regression models. The first model examined the association between use of various substances, including CM, and violent recidivism. The second model examined the association between CM use, polysubstance use, presence of a psychiatric diagnosis, history of violence, and subsequent violent recidivism.

Nagelkerke's $R^2$ for model 1 was $R^2 = 0.02$ and for model 2 $R^2 = 0.17$.

The associations between substance use and violent recidivism are presented in Table 3. None of the variables entered into the model was significantly associated with violent recidivism (all p > .05).

Table 4 presents the associations between CM use, polysubstance use, the presence of a psychiatric diagnosis, a history of violence, and subsequent violent recidivism. Of the terms included in the model, only the presence of a psychiatric diagnosis was significantly associated with violent recidivism (AOR = 1.8, 95% CI 1.0–3.4). Four interaction terms were also included in the model, however none of them was significantly associated with the outcome.

Given that the variable for psychiatric diagnosis was considerably heterogeneous in terms of diagnoses included, a sensitivity analysis was conducted with four homogenous sub-groups

**Table 2. Correlates of crystalline methamphetamine use.**

| | No CM Use (N = 123) | | | CM Use (N = 75) | | | OR (95%CI) | AOR (95%CI)[a] |
|---|---|---|---|---|---|---|---|---|
| | n | % | 95% CI | n | % | 95% CI | | |
| Age (per year older) | - | - | - | - | - | - | 1.3 (1.1, 1.5)** | 1.3 (1.1, 1.6)** |
| Female | 34 | 28 | 20, 37 | 1 | 1 | 1<, 8 | ref | ref |
| Male | 89 | 72 | 63, 80 | 74 | 99 | 92, >99 | 28.3 (3.8, 211.5)*** | 25.5 (5.6, 117.2)*** |
| No polysubstance use | 95 | 77 | 69, 84 | 38 | 51 | 39, 62 | ref | ref |
| Polysubstance use[a] | 28 | 23 | 16, 31 | 37 | 49 | 38, 61 | 3.3 (1.8, 6.1)*** | 4.7 (2.2, 10.1)*** |
| Frequency of alcohol use (days per month) | - | - | - | - | - | - | 1.0 (1.0, 1.2) | 1.0 (0.9, 1.0) |
| No psychiatric diagnosis | 72 | 59 | 49, 67 | 53 | 71 | 59, 80 | ref | ref |
| Psychiatric diagnosis | 51 | 41 | 33, 51 | 22 | 29 | 20, 41 | 0.6 (0.3, 1.1) | 0.6 (0.3, 1.2) |
| Low community disorganisation | 47 | 38 | 30, 47 | 16 | 21 | 13, 33 | ref | ref |
| Mod. community disorganisation | 38 | 31 | 23, 40 | 22 | 29 | 20, 41 | 1.7 (0.8, 3.7) | 2.3 (0.9, 5.9) |
| High community disorganisation | 37 | 31 | 22, 39 | 37 | 49 | 38, 61 | 2.9 (1.4, 5.9)** | 6.3 (2.4, 16.6)*** |
| Low peer delinquency | 6 | 5 | 2, 11 | 4 | 5 | 2, 13 | ref | ref |
| Mod. peer delinquency | 37 | 30 | 22, 39 | 12 | 16 | 9, 27 | 0.5 (0.1, 2.0) | 0.4 (0.1, 1.7) |
| High peer delinquency | 80 | 65 | 56, 73 | 59 | 79 | 67, 87 | 1.1 (0.3, 4.1) | 0.5 (0.1, 2.2) |
| Low poor parental management | 20 | 16 | 10, 24 | 11 | 15 | 8, 25 | ref | ref |
| Mod. poor parental management | 39 | 32 | 24, 41 | 22 | 29 | 20, 41 | 1.0 (0.4, 2.5) | 0.9 (0.3, 2.7) |
| High poor parental management | 64 | 52 | 43, 61 | 42 | 56 | 44, 67 | 1.2 (0.5, 2.7) | 1.1 (0.4, 3.0) |

[a]Polysubstance use did not include CM use

*p < .05

**p < .01

***p < .001

**Table 3. Association between substance use and subsequent violent recidivism.**

|  | No Recidivism (N = 58) | | | Recidivism (N = 102) | | | | |
|---|---|---|---|---|---|---|---|---|
|  | N | % | (95% CI) | n | % | (95% CI) | OR (95%CI) | AOR (95%CI)[a] |
| No CM | 36 | 62 | 48, 74 | 62 | 61 | 51, 70 | ref | ref |
| CM | 22 | 38 | 26, 52 | 40 | 39 | 30, 49 | 1.1 (0.5, 2.0) | 1.0 (0.7, 1.5) |
| No Cannabis | 7 | 12 | 5, 24 | 15 | 15 | 9, 23 | ref | ref |
| Cannabis | 51 | 88 | 76, 95 | 87 | 85 | 77, 91 | 0.8 (0.3, 2.1) | 1.1 (0.7, 1.6) |
| No Sedatives | 52 | 90 | 78, 96 | 80 | 78 | 69, 86 | ref | ref |
| Sedatives | 6 | 10 | 4, 22 | 22 | 22 | 14, 31 | 2.4 (0.9, 6.3) | 1.2 (0.8, 1.8) |
| No Cocaine | 50 | 86 | 74, 93 | 94 | 92 | 85, 96 | ref | ref |
| Cocaine | 8 | 14 | 7, 26 | 8 | 8 | 4, 15 | 0.5 (0.2, 1.5) | 0.9 (0.6, 1.4) |
| No Opioid use | 51 | 88 | 76, 95 | 83 | 81 | 72, 88 | ref | ref |
| Opioid use | 7 | 12 | 5, 24 | 19 | 19 | 12, 28 | 1.7 (0.7, 4.2) | 1.1 (0.7, 1.7) |
| No hallucinogens | 53 | 91 | 80, 97 | 92 | 90 | 82, 95 | ref | ref |
| Hallucinogens | 5 | 9 | 3, 20 | 10 | 10 | 5, 18 | 1.2 (0.4, 3.6) | 1.0 (0.6, 1.5) |
| No solvents | 56 | 97 | 87, 99 | 89 | 87 | 79, 93 | ref | ref |
| Solvents | 2 | 3 | 1, 13 | 13 | 13 | 7, 21 | 4.1 (0.9, 18.9) | 1.2 (0.8, 1.8) |
| Frequency of alcohol use (days per month) | - | - | - | - | - | - | 1.0 (0.9, 1.1) | 1.0 (0.9, 1.0) |

[a] Odds ratios are adjusted for all other variables listed in the model.

of diagnoses–mood disorders, behavioural disorders, anxiety disorders, trauma related disorder. Three of the four categories examined for participants who re-offended—mood disorders (21.4%), behavioural disorders (28.2%), and trauma related disorder (23.3%)—were present at similar rates among those who were subsequently charged with a violent offence in the follow up period. Diagnoses of anxiety disorders were present at a slightly lower rate among participants who re-offended (17.5%).

**Table 4. Associations between mental health, polysubstance use history of violence, CM use and violent recidivism.**

|  | No offending (N = 58) | | | offending (N = 102) | | | | |
|---|---|---|---|---|---|---|---|---|
|  | N | % | (95% CI) | N | % | (95% CI) | OR (95%CI) | AOR (95%CI)[b] |
| No CM use | 36 | 62 | 48, 74 | 62 | 61 | 51, 70 | ref | ref |
| CM use | 22 | 38 | 26, 52 | 40 | 39 | 30, 49 | 1.1 (0.5, 2.0) | 1.1 (0.6, 2.3) |
| No polysubstance use | 42 | 72 | 59, 83 | 63 | 62 | 52, 71 | ref | ref |
| Polysubstance use[a] | 16 | 28 | 17, 41 | 39 | 38 | 29, 48 | 1.6 (0.8, 3.3) | 1.3 (0.6, 2.6) |
| Low/moderate history of violence | 12 | 21 | 12, 34 | 21 | 21 | 13, 30 | ref | ref |
| High history of violence | 46 | 79 | 66, 88 | 81 | 79 | 70, 87 | 1.0 (0.5, 2.2) | 0.9 (0.5, 1.7) |
| No psychiatric diagnosis | 45 | 78 | 64, 87 | 52 | 51 | 41, 61 | ref | ref |
| Psychiatric diagnosis | 13 | 22 | 13, 36 | 50 | 49 | 39, 59 | 3.3 (1.6, 6.9)*** | 1.8 (1.0, 3.4)* |
| CM use * Psychiatric diagnosis | - | - | - | - | - | - | - | 1.1 (0.4, 3.1) |
| CM use * History of violence | - | - | - | - | - | - | - | 0.8 (0.4, 1.9) |
| Polysubstance use * History of violence | - | - | - | - | - | - | - | 0.9 (0.4, 2.2) |
| Polysubstance use * Psychiatric diagnosis | - | - | - | - | - | - | - | 2.1 (0.6, 7.1) |

[a]Polysubstance use did not include CM use

[b] Odds ratios are adjusted for all variables listed in the model

*p < .05

**p < .01

***p < .001

## Discussion

This study is the first internationally to examine the prevalence and correlates of crystalline methamphetamine (CM) use in a sample of young people released from juvenile detention. We also examined the association between CM use and violent recidivism.

### Prevalence of CM use

More than one third of participants reported lifetime CM use. Rates of substance use are often much higher in youth justice populations[26,27], but the high prevalence of CM use in this sample compared to the general Australian population (estimates range from 3–4%, [3,10]) is particularly striking. The lifetime prevalence of use of other substances (e.g., cannabis, 86%) in this sample was also considerably higher than in the general population. One third of the sample had previously used two or more of the substances we examined and, of these, more than half (57%) had also used CM. We were unable to determine whether CM use preceded or succeeded the use of other illicit drugs. Given that in the general population the mean age for first use of cannabis is typically lower than mean age for first use of CM [3], it is likely that CM was not the first substance ever used by participants. Previous reports indicate that CM users often consume additional drugs and engage in risky drinking behaviour [3]. The high rates of polysubstance use reported in our study highlight a major area of therapeutic need for youth custodial populations.

### Correlates of CM use

Consistent with evidence from the general population that age at first use of methamphetamines is higher than for some other substances such as cannabis [3], we found that older participants were more likely to report a history of CM use. We also found that young males were more than 25 times more likely than young females to report CM use, suggesting a gendered nature of CM use in this population. Our findings are consistent with prior research with young people (aged 15–29) in Australia, which found that both being male and older age were associated with CM use[10]. Similarly, an Australian study investigating the characteristics of CM users presenting to a hospital emergency department found the cohort to be predominantly male and between the ages of 26–30 years[8].

We also found that polysubstance use was associated with CM use. This finding was anticipated given that more than half of the participants who reported lifetime CM use also reported using other substances. CM, at least for members of this cohort, is perhaps seldom used in isolation and may be consumed by those who already have histories of illicit drug use. A recent Australian study of 564 people who inject drugs found that only a small percentage of the cohort reported CM as the substance most used over the past month[41].

Participants who reported living in a community with high levels of disorganisation were almost six times more likely than participants from communities with low levels of disorganisation to have used CM. The SAVRY operationalisation of this item refers to communities with high rates of crime, poverty and violence. Such communities often have a higher availability of illicit drugs. Moreover, research from Australia suggests a greater of availability of CM in the community in recent years[42]. Associations have been drawn between neighbourhood factors (i.e., disadvantage, high-crime rates, instability, disorganization, perceived drug selling) and youth substance use [43–47]. Having friends who use illicit substances is also linked with substance use[3,48]. The associations found between community disorganisation and substance use among justice involved young people point to the importance of broader community based strategies to reduce CM use among this vulnerable population following their release from detention.

## CM use and violent recidivism

Over a span of 13 months, more than half of the sample was charged with at least one violent offence. We found no association between use of CM or any other substance and violent recidivism; however, a history of psychiatric diagnosis was associated with violent recidivism. The association between mental disorder and recidivism is consistent with previous research on youth custodial populations, with several studies reporting high rates of mental disorder among this population [29–31,49] and an association with future recidivism [50].

Research examining the relationship between psychiatric diagnoses and violent recidivism in this population has pointed to the fact that common symptoms of psychiatric illnesses such as substance use, impulsivity and irritability (as related to substance use disorders and attention deficit disorders for example) are also common risk factors for violent recidivism[51,52]. While a causal relationship has not been established between symptomology and violent recidivism, the relationship appears to suggest that treatment of mental health symptoms could have utility in reducing violent recidivism. These findings highlight the importance of adequate investment in evidence-based, continuous and coordinated mental health care for young people who cycle through youth detention, as a means of reducing violent recidivism.

## Limitations

This is the first study to examine correlates of CM use among detained youth, however it had a number of limitations. Firstly, we examined only lifetime use of CM and did not include measurements of frequency of use, use within a specified period (e.g., past 12 months), or estimates of typical quantity used. However, given the relatively young age of the sample, it is likely that only a minority had ceased using substances for which they reported lifetime use. Furthermore, the study relied entirely on self-report measurements of CM use. While self-report is a useful method of measuring CM use, it is vulnerable to failures of memory and reticence to answer sensitive questions relating to drug use. Ideally, future research would incorporate data from other sources (e.g., linked drug treatment or hospital records, urinalysis) to triangulate measurement of CM use in this population. Finally, the modest sample size limited statistical power, and although we had a high recruitment fraction it was not possible to formally assess sample representativeness. Our estimates of prevalence and association will require replication in larger samples.

## Implications and conclusions

Our study provides further evidence that rates of lifetime substance use are substantially higher among young people in detention than in the general population [26,27]. Given the established associations between substance use and adverse psychiatric and physical health outcomes, there is a clear need for processes to identify and support detained youth who have or are at-risk of developing substance use problems. Identification should involve not only drug testing, but also the use of interviews and screening measures that can inform treatment responses, both in detention and after these young people return to the community.

Although there is good evidence that CM use is associated with poor health outcomes [4–9], in this study neither CM use nor use of other substances was associated with violent recidivism. The assessment of violence risk is an area of increasing importance for health professionals operating in youth detention settings, and also for corrections staff. While our findings suggest that CM use was not associated with violent recidivism, the measurement of CM use was not sensitive to the frequency and extent of CM use. It is quite possible that an association between CM use and violent recidivism exists when taking into account a more nuanced measurement. More detailed assessment of the relationship between CM use and violent

recidivism, taking into account factors like method of administration, frequency of use and average amount of use may uncover important relationships that ought to be considered in determining which young people should be selected for treatment designed to reduce risk of violent recidivism.

## Author Contributions

**Conceptualization:** Benjamin Spivak, Stephane Shepherd, Rohan Borschmann, Stuart A. Kinner, Henning Hachtel.

**Data curation:** Henning Hachtel.

**Formal analysis:** Benjamin Spivak, Stephane Shepherd, Henning Hachtel.

**Methodology:** Benjamin Spivak, Henning Hachtel.

**Project administration:** Stephane Shepherd.

**Writing – original draft:** Benjamin Spivak, Stephane Shepherd.

**Writing – review & editing:** Benjamin Spivak, Stephane Shepherd, Rohan Borschmann, Stuart A. Kinner, James R. P. Ogloff, Henning Hachtel.

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
