## [Decision Letter · Decision Letter 0]

28 Jan 2020

PONE-D-19-31703

Crystalline methamphetamine (ice) use prior to youth detention: A forensic concern or a public health issue?

PLOS ONE

Dear Dr Shepherd,

Thank you for submitting your manuscript to PLOS ONE. After careful consideration, we feel that it has merit but does not fully meet PLOS ONE’s publication criteria as it currently stands. Therefore, we invite you to submit a revised version of the manuscript that addresses the points raised during the review process.

I have read the paper and agree with the reviewers that it makes a useful contribution to limited evidence in this important area. The study is methodological sound with the use of linkage data however I felt it was lacking in detail in several areas (e.g., in relation to the databases used). I believe addressing this would strengthen the paper given the international readership of this journal.

The abstract needs to be clearer in the results section that CM was not associated with re-offending (i.,e., post-release) as this was one of the 2 main aims of the study.

Please add ethics approval reference

More information is needed on how mental health diagnoses are made, i.e., how they get recorded in RAPID. It may also be worth noting in the limitations section that these may an underestimate if they had mental health issues that had gone undetected/not diagnosed. Similarly how were de-identified records examined with Victoria Police) - was there a statistical linkage key used?

Please clarify why violent offending data time period was so variable - up to 18 months is reported in the method but the range indicated in the results sections suggests it was as early as 1-month for some - surely this would limit the chances of there being a reported offence?

The discussion could go further in unpacking the demonstrated relationship between psychiatric diagnosis and reoffending (e.g., poor impulse control, lack of insight etc).

I also think the paper would benefit from my consistent methodology - (recidivism, re-offending, future violent criminal offending).

We would appreciate receiving your revised manuscript by Mar 13 2020 11:59PM. To enhance the reproducibility of your results, we recommend that if applicable you deposit your laboratory protocols in protocols.io, where a protocol can be assigned its own identifier (DOI) such that it can be cited independently in the future. For instructions see: http://journals.plos.org/plosone/s/submission-guidelines#loc-laboratory-protocols

We look forward to receiving your revised manuscript.

Kind regards,

Victoria Manning

Academic Editor

PLOS ONE

2. Please provide additional information regarding the considerations made for the prisoners included in this study.

For instance, please discuss whether participants were able to opt out of the study and whether individuals who did not participate receive the same treatment offered to participants.

Please specifically clarify whether consent procedures were approved by IRB

4. Thank you for stating the following in the Declarations Section of your manuscript:

'Funding

This research was funded by the Australian Research Council – grant number DP1095697'

'This study was funded by the Australian Research Council (www.arc.gov.au). The funders had no role in study design, data collection and analysis, decision to publish, or preparation of the manuscript.'

Please provide an amended Funding Statement that declares *all* the funding or sources of support received during this specific study (whether external or internal to your organization) as detailed online in our guide for authors at http://journals.plos.org/plosone/s/submit-now Please state what role the funders took in the study.  If any authors received a salary from any of your funders, please state which authors and which funder. If the funders had no role, please state: "The funders had no role in study design, data collection and analysis, decision to publish, or preparation of the manuscript."

5. Your ethics statement must appear in the Methods section of your manuscript. If your ethics statement is written in any section besides the Methods, please move it to the Methods section and delete it from any other section. Please also ensure that your ethics statement is included in your manuscript, as the ethics section of your online submission will not be published alongside your manuscript.

Reviewers' comments:

Reviewer's Responses to Questions

**Comments to the Author**

1. Is the manuscript technically sound, and do the data support the conclusions?

Reviewer #1: Partly

Reviewer #2: Yes

2. Has the statistical analysis been performed appropriately and rigorously? 

Reviewer #1: No

Reviewer #2: Yes

3. Have the authors made all data underlying the findings in their manuscript fully available?

Reviewer #1: No

Reviewer #2: No

4. Is the manuscript presented in an intelligible fashion and written in standard English?

Reviewer #1: Yes

Reviewer #2: Yes

5. Review Comments to the Author

Reviewer #1: This study sought to identify (a) the lifetime prevalence of crystalline methamphetamine (CM) use, (b) the correlates of CM use, and (c) the association between CM use and future violent criminal offending among a sample of young people in detention. The participants were 202 young people in youth detention in the state of Victoria, Australia. Results revealed that more than one-third of the sample (38%) reported lifetime CM use, with older age, male gender, polysubstance use, and high levels of community disorganization being associated with CM use. The presence of a psychiatric diagnosis over the lifetime was not significantly associated with CM use, and CM use was not significantly associated with violent offending.

Given the media attention focused on the perceived relationship between CM use and offending, particularly violent offending, an empirical investigation of the relationship has obvious scientific merit. The authors should therefore be commended for undertaking this research. This study has several strengths, including a convincing rationale, a clear design and methodology, straightforward statistical analyses, and a solid synthesis of the results with the existing literature. I also appreciated the use of precise language that did not imply causality. Despite these strengths, there are several notable methodological limitations and several aspects of the manuscript in need of attention.

(1) The likely lack of generalizability of the study findings is a significant concern, particularly in light of the stated goals of the study. The sample size is relatively small (N = 202), with an even smaller subset (n = 76) who reported lifetime CM use. Also, the sample was drawn from two youth justice facilities in one Australian state. The small, geographically limited sample raises concerns that the sample may not be representative. The manuscript appropriately acknowledged that “it was not possible to formally assess sample representativeness” (see Limitations), but it is difficult to overlook this limitation. A stated study goal was to assess lifetime prevalence use of CM, but to do so in a meaningful and defensible manner requires a larger, representative sample.

(2) The manuscript indicates that informed consent was obtained from all participants over age 18, and that consent for participants under age 18 fell within the “mature minor” concept. However, there is no indication of whether anyone refused to provide informed consent, or whether any of the participants under age 18 were determined to lack the capacity to understand the nature of the study.

(3) A related concern is the lack of information regarding the recruitment fraction. The manuscript states that the authors achieved a “high recruitment fraction” (see Limitations), but no other information is provided. As such, it is not possible to assess whether the sample size of 202 is a small, moderate, or large sample of the population of interest. Providing the census of the two facilities would be informative.

(4) The use of a binary variable for CM use (0 = no CM use; 1 = lifetime CM use) lacks sufficient specificity and sensitivity. When addressing this limitation, the manuscript stated: “Firstly, we examined only lifetime use of CM and did not include measurements of frequency of use, use within a specified period (e.g., past 12 months), or estimates of typical quantity used” (see Limitations). Acknowledging this limitation is laudable, but it is difficult to overlook. Given the binary nature of this variable, participants who used CM once are grouped with participants who used CM many times. Such a crude classification system does not acknowledge the differences that (likely) exist between those with limited CM use and those with extensive CM use; grouping them together adds too much noise to the data, and the study results pertaining to CM use, the relationship between CM use and other drug use, and the relationship between CM use and violence become less defensible and less meaningful. For example, concluding that more than one-third of the sample reported lifetime CM use could be misleading because we do not know what proportion of those participants used CM once versus many times. To be clear, using CM once is indeed lifetime CM use, but it may not make sense (given what we know about episodic versus chronic CM use) to group one-time CM users with chronic CM users.

(5) There is a similar sensitivity problem related to the definition of violent offending. The manuscript defined violent recidivism as “any personal injury transgression that led to a police charge,” which included “acts intended to cause or threaten to cause physical harm to a victim.” There are two concerns. First, defining recidivism based on charges, as opposed to arrests or convictions, would benefit from being justified. Second, combining acts with threats introduces noise into the data because there are differences between acts of violence and threats of violence. Some justification for combining acts and threats would strengthen the manuscript.

(6) The manuscript does not include an a priori statistical power analysis. The manuscript mentions that the “modest sample size limited statistical power,” but (a) it is not clear how the authors know that statistical power was in fact limited, and (b) the lack of a power analysis (and potentially low power) raises concerns about Type II errors.

(7) The manuscript does not include any testable hypotheses.

(8) To support the assertion that the adolescent brain is still developing, the manuscript cites an article from 1999. The 1999 article is certainly a solid reference, but perhaps a more recent reference that reflects the current brain science would be more informative.

Thank you for the opportunity to review this manuscript.

Reviewer #2: This study reports on the association between a lifetime history of methamphetamine use and violent offending in a prospective study of people in youth detention. No previous studies have examined this relationship longitudinally in this population (see recent systematic review by McKetin et al in eClinicalMedicine 16: 87-91).

The main weakness of the present study is the exposure measure (any lifetime use) lacks the ability to differentiate between people who have used frequently and those with only occasional use. Its main strength is the used of linkage data for the outcome. The methods and results are clearly written, and the analytic approach appears sound.

Comments are minor:

1. Section 2.1 needs more information about the sampling frame & sampling strategy- how were participants selected?

2. Regarding the associations between having a psychiatric diagnosis and violent recidivism, is it possible this could be explained by reverse causation- that is, violent offences led to psychiatric assessment which then resulted in a diagnosis being recorded?

3. The comment “Our findings suggest that CM use on its own is unlikely to be an important consideration” etc over-reaches what this study shows, since it’s possible the association with violence is confined to people who have used CM frequently (see Foulds et al- “Methamphetamine use and violence: findings from a longitudinal birth cohort”- Drug and Alcohol Dependence, in press). Thus an alternative conclusion would be that more detailed assessment of CM use- ie frequency, route of administration and presence of substance use disorder- is necessary to aid treatment selection. Though it is noted that no evidence-based treatments specific to methamphetamine use disorder exist at this time.

4. In Table 2, 3 and 4 the inclusion of rows for the reference categories (eg “no cannabis”) is perhaps not necessary. Consider omitting the reference category rows unless these are essential to the data presentation.

6. PLOS authors have the option to publish the peer review history of their article (what does this mean?). If published, this will include your full peer review and any attached files.

Reviewer #1: No

Reviewer #2: Yes: James Foulds

---

## [Author Response · Author response to Decision Letter 0]

3 Feb 2020

Editors’ comments

The abstract needs to be clearer in the results section that CM was not associated with re-offending (i.,e., post-release) as this was one of the 2 main aims of the study.

Thank you for pointing out this omission. We have revised the abstract to include this information.

Please add ethics approval reference

The ethics approval number has now been added to the method section

More information is needed on how mental health diagnoses are made, i.e., how they get recorded in RAPID. It may also be worth noting in the limitations section that these may an underestimate if they had mental health issues that had gone undetected/not diagnosed. Similarly how were de-identified records examined with Victoria Police) - was there a statistical linkage key used?

We have now added information about how diagnoses are recorded into RAPID in the method section. In regards to the de-identified records, yes, a linkage key was used. We have now also incorporated this information into the method section.

Please clarify why violent offending data time period was so variable - up to 18 months is reported in the method but the range indicated in the results sections suggests it was as early as 1-month for some - surely this would limit the chances of there being a reported offence?

This is a typographical error. The values reported actually refer to time to re-offence. We have now amended the text to reflect this. We have also included the appropriate information about follow up time with 6 months minimum for the entire sample up to 18 months.

The discussion could go further in unpacking the demonstrated relationship between psychiatric diagnosis and reoffending (e.g., poor impulse control, lack of insight etc).

We agree and have now included additional information about the relationship between psychiatric diagnoses and re-offending in the discussion section.

I also think the paper would benefit from my consistent methodology - (recidivism, re-offending, future violent criminal offending).

We assume this point relates to terminology. We agree and have made the terminology in relation to violent recidivism consistent throughout the manuscript

Reviewer 1 comments

(1) The likely lack of generalizability of the study findings is a significant concern, particularly in light of the stated goals of the study. The sample size is relatively small (N = 202), with an even smaller subset (n = 76) who reported lifetime CM use. Also, the sample was drawn from two youth justice facilities in one Australian state. The small, geographically limited sample raises concerns that the sample may not be representative. The manuscript appropriately acknowledged that “it was not possible to formally assess sample representativeness” (see Limitations), but it is difficult to overlook this limitation. A stated study goal was to assess lifetime prevalence use of CM, but to do so in a meaningful and defensible manner requires a larger, representative sample.

The determination of what constitutes a ‘small sample’ is dependent on what population we are hoping to generalise to. The average number of youths in detention in the whole of Australia varies from around 800 youths to 1000 depending on the time measured. Our sample included 202 participants (roughly 20% of the national amount). We do concede that the sample may not be representative, as the sample was drawn from a single Australian state and may differ demographically from populations in other states, we have tried to point this out to readers in the limitations section. We feel that the results are still meaningful for a number of reasons. Firstly, this is a novel area of research with no existing published studies on estimated lifetime CM use. Given this dearth of information, even a biased estimate is preferable to no estimate especially given that readers are provided with enough contextual information in both the method and limitations section to avoid overconfidence in the estimates provided. The alternative to the estimates we provide is to rely on pure guesswork. Secondly, estimates from smaller sample sizes are still meaningful provided that too much confidence is not placed on the results of statistical analyses. As has been pointed out numerous times, overall statistical estimates (say from meta-analytic methods) from any field are likely to be biased when small n studies are not published and available to base conclusions on.

(2) The manuscript indicates that informed consent was obtained from all participants over age 18, and that consent for participants under age 18 fell within the “mature minor” concept. However, there is no indication of whether anyone refused to provide informed consent, or whether any of the participants under age 18 were determined to lack the capacity to understand the nature of the study.

Thank you for identifying this omission. We have corrected the method to include a statement that indicates that all participants were determined to have capacity to understand the nature of the study and that no participants refused to provide informed consent.

(3) A related concern is the lack of information regarding the recruitment fraction. The manuscript states that the authors achieved a “high recruitment fraction” (see Limitations), but no other information is provided. As such, it is not possible to assess whether the sample size of 202 is a small, moderate, or large sample of the population of interest. Providing the census of the two facilities would be informative.

We agree that this is information that should have been included. We have amended the participant section to provide population level estimates from Government agencies to provide readers with the necessary information to gauge the sample size relative to population.

(4) The use of a binary variable for CM use (0 = no CM use; 1 = lifetime CM use) lacks sufficient specificity and sensitivity. When addressing this limitation, the manuscript stated: “Firstly, we examined only lifetime use of CM and did not include measurements of frequency of use, use within a specified period (e.g., past 12 months), or estimates of typical quantity used” (see Limitations). Acknowledging this limitation is laudable, but it is difficult to overlook. Given the binary nature of this variable, participants who used CM once are grouped with participants who used CM many times. Such a crude classification system does not acknowledge the differences that (likely) exist between those with limited CM use and those with extensive CM use; grouping them together adds too much noise to the data, and the study results pertaining to CM use, the relationship between CM use and other drug use, and the relationship between CM use and violence become less defensible and less meaningful. For example, concluding that more than one-third of the sample reported lifetime CM use could be misleading because we do not know what proportion of those participants used CM once versus many times. To be clear, using CM once is indeed lifetime CM use, but it may not make sense (given what we know about episodic versus chronic CM use) to group one-time CM users with chronic CM users.

We agree that this is an important limitation and largely agree with the reviewer that the relationship between CM use and outcomes would be more nuanced with further information on the extent of CM use among the sample. We have amended our discussion to more strongly point out this limitation. We do feel however, that the results represent a limited but important first step in an area where there has not been any published research and provide a broad estimate of the relationship between CM use, substance use and offending. 

(5) There is a similar sensitivity problem related to the definition of violent offending. The manuscript defined violent recidivism as “any personal injury transgression that led to a police charge,” which included “acts intended to cause or threaten to cause physical harm to a victim.” There are two concerns. First, defining recidivism based on charges, as opposed to arrests or convictions, would benefit from being justified. Second, combining acts with threats introduces noise into the data because there are differences between acts of violence and threats of violence. Some justification for combining acts and threats would strengthen the manuscript.

We have provided further information to justify the definition of violent recidivism within the manuscript. 

The use of charges as our measure of recidivism was chosen for a number of reasons. Firstly, the use of arrests as an outcome measure would underestimate the amount of violent recidivism given that not all charges for violence result in arrest. The use of charges was also preferred to convictions given the relatively small follow up period in the current study (6-18 months). The time needed for a case to proceed from incident to charge to trial to conviction would mean that the base rate for the outcome in the time period examined would be relatively low. Furthermore the use of conviction is likely to also underestimate offending given that mistrials and prosecution discretion are likely to artificially reduce the rate of violent recidivism in a way that is not necessarily accurate.

The grouping of threats alongside physical acts of violence is now commonplace. For example the World Health Organisation defines violence as “"the intentional use of physical force or power, threatened or actual, against oneself, another person, or against a group or community..." (Krug et al., 2002). We have amended the method to make the reason for the grouping of the two forms of violence clearer.

(6) The manuscript does not include an a priori statistical power analysis. The manuscript mentions that the “modest sample size limited statistical power,” but (a) it is not clear how the authors know that statistical power was in fact limited, and (b) the lack of a power analysis (and potentially low power) raises concerns about Type II errors.

The use of an a priori power analysis was not possible for the current analysis given that the data had already been collected prior to analysis. The reviewer is correct that we could not possibly ‘know’ that statistical power was limited, however we suspect that it is limited given that guidelines from simulation studies and textbooks suggest that we would require more data to estimate effects at sufficient power. For example, Peduzzi et al. (1996) offers the following formula for estimating sample size: N = (10*K)/P, where K is the number of covariates and P is the proportion of outcome events in the sample. With our data (which is not necessarily representative of the population) we can use this rule of thumb to estimate in the case of model 2 that we would need 211 cases whereas we had in reality 198. We agree about the possibility of type II errors and we have now made this point in the discussion.

(7) The manuscript does not include any testable hypotheses

The lack of hypotheses was intentional. Given the novelty of research in this area, we did not feel comfortable specifying hypotheses.

(8) To support the assertion that the adolescent brain is still developing, the manuscript cites an article from 1999. The 1999 article is certainly a solid reference, but perhaps a more recent reference that reflects the current brain science would be more informative

We have now included a reference to more recent research in brain science which supports the assertion that the adolescent brain is still developing.

Reviewer 2 comments

1. Section 2.1 needs more information about the sampling frame & sampling strategy- how were participants selected?

All young people (remanded and sentenced) in the two custodial centres were eligible to take part in the study so long as they were able to comprehend the purpose and nature of the study and then provide consent. Participants were eligible to participate if they were English speaking and able to comprehend the participatory explanation form. Detainees were not approached if Justice Centre staff deemed them to be of unstable mood or likely to exhibit extreme aggressive behavior if interviewed. The above information has now been incorporated into the paper.

2. Regarding the associations between having a psychiatric diagnosis and violent recidivism, is it possible this could be explained by reverse causation- that is, violent offences led to psychiatric assessment which then resulted in a diagnosis being recorded?

This is not possible in the current study given that the psychiatric diagnoses were recorded prior to the recidivism. 

3. The comment “Our findings suggest that CM use on its own is unlikely to be an important consideration” etc over-reaches what this study shows, since it’s possible the association with violence is confined to people who have used CM frequently (see Foulds et al--Methamphetamine use and violence: findings from a longitudinal birth cohort”- Drug and Alcohol Dependence, in press). Thus an alternative conclusion would be that more detailed assessment of CM use- ie frequency, route of administration and presence of substance use disorder- is necessary to aid treatment selection. Though it is noted that no evidence-based treatments specific to methamphetamine use disorder exist at this time.

We wholeheartedly agree with this suggestion. We have amended the discussion to reflect this possibility.

4. In Table 2, 3 and 4 the inclusion of rows for the reference categories (eg “no cannabis”) is perhaps not necessary. Consider omitting the reference category rows unless these are essential to the data presentation.

We feel that the rows for the reference categories contain important descriptive information in terms of the numbers of individuals who did and did not have the outcome of interest (e.g. violence) in the reference category. This is potentially useful for those who may wish to incorporate the study in a future systematic review.

References

Krug, E.G., Dahlberg, L.L., Mercy J.A., Zwi, A.B., & Lozano, R. (2002).World report on violence and health. Geneva: World Health Organization

Peduzzi, P, Concato, J., Kemper, E., Holford, T.R., & Feinstein, A.R. (1996). A simulation study of the number of events per variable in logistic regression analysis. Journal of Clinical Epidemiology, 49, 1373-1379

---

## [Editor Report · Decision Letter 1]

27 Mar 2020

Crystalline methamphetamine (ice) use prior to youth detention: A forensic concern or a public health issue?

PONE-D-19-31703R1

Dear Dr. Shepherd

We are pleased to inform you that your manuscript has been judged scientifically suitable for publication and will be formally accepted for publication once it complies with all outstanding technical requirements.

With kind regards,

Gurudutt Pendyala, Ph.D.

Academic Editor

PLOS ONE
---

## [Editor Report · Acceptance letter]

12 Feb 2020

PONE-D-19-31703R1 

Crystalline methamphetamine (ice) use prior to youth detention: A forensic concern or a public health issue? 

Dear Dr. Shepherd:

I am pleased to inform you that your manuscript has been deemed suitable for publication in PLOS ONE. Congratulations! Your manuscript is now with our production department. 

With kind regards,

on behalf of

Dr. Victoria Manning 

Academic Editor

PLOS ONE